# Transpiration efficiency variations in the pearl millet reference collection PMiGAP

**Laura Grégoire[1], Jana Kholova[2,3], Rakesh Srivastava[2], Charles Thomas Hash[2], Yves Vigouroux[1], Vincent Vadez[1,2,4]***

1 Diversité, Adaptation, Développement des Plantes (DIADE), University of Montpellier, Institut de Recherche pour le Développement (IRD), Montpellier, France, 2 International Crop Research Institute in Semi-Arid Tropics (ICRISAT), Hyderabad, India, 3 Department of Information Technologies, Faculty of Economics and Management, Czech University of Life Sciences Prague, Prague, Czech Republic, 4 Centre d'étude Régional Pour l'amélioration de l'adaptation à la Sécheresse (CERAAS), Thiès, Sénégal

* vincent.vadez@ird.fr

**Data Availability Statement:** All relevant data for this study are publicly available from the DataSuds repository (https://doi.org/10.23708/2VJ0TQ).

**Funding:** LG and VV was supported by the Make Our Planet Great Again (MOPGA) ICARUS project

## Abstract

Transpiration efficiency (TE), the biomass produced per unit of water transpired, is a key trait for crop performance under limited water. As water becomes scarce, increasing TE would contribute to increase crop drought tolerance. This study is a first step to explore pearl millet genotypic variability for TE on a large and representative diversity panel. We analyzed TE on 537 pearl millet genotypes, including inbred lines, test-cross hybrids, and hybrids bred for different agroecological zones. Three lysimeter trials were conducted in 2012, 2013 and 2015, to assess TE both under well-watered and terminal-water stress conditions. We recorded grain yield to assess its relationship with TE. Up to two-fold variation for TE was observed over the accessions used. Mean TE varied between inbred and test-cross hybrids, across years and was slightly higher under water stress. TE also differed among hybrids developed for three agroecological zones, being higher in hybrids bred for the wetter zone, underlining the importance of selecting germplasm according to the target area. Environmental conditions triggered large Genotype x Environment (GxE) interactions, although TE showed some high heritability. Transpiration efficiency was the second contributor to grain yield after harvest index, highlighting the importance of integrating it into pearl millet breeding programs. Future research on TE in pearl millet should focus (i) on investigating the causes of its plasticity i.e. the GxE interaction (ii) on studying its genetic basis and its association with other important physiological traits.

## Introduction

Pearl millet [Pennisetum glaucum (L.) R. Br.]is one of the most drought tolerant cereals. It is grown as fodder and rainfed cereal in arid and semi-arid regions, where annual rainfall is low and intermittent. Despite its drought tolerance, this crop is nonetheless subject to abiotic stresses, such as terminal drought stress. This translates into a lack of available water during the terminal and crucial phenology stages (from flowering to grain filling) which is linked to an interruption in rainfall towards the end of the season and a progressive depletion of the soil

(Improve Crops in Arid Regions and future climates) funded by the Agence Nationale de la Recherche (ANR, grant ANR-17-MPGA-0011), by the Occitanie Region through a financial contribution to grant ANR-17-MPGA-0011, and by Montpellier University of Excellence (I-Site MUSE). JK was supported by the Internal grant agency of the Faculty of Economics and Management, Czech University of Life Sciences Prague, grant no. 2023B0005 (Oborově zaměřené datové modely pro podporu iniciativy Open Science a principu FAIR). the funders had no role in study design, data collection and analysis, decision to publish, or preparation of the manuscript.

**Competing interests:** The authors have declared that no competing interests exist.

water. The consequence of that frequent stress type is that it reduces the filling of grains and reduces grain number, grain quality (size in particular) and total grain yield. It has the most radical impact on pearl millet grain and forage yield, as well as yield stability, with yield losses of around 55–67% recorded due to terminal drought stress [1–5]. In this context, having higher transpiration efficiency (TE), i.e. the amount of biomass produced per unit of water transpired, could be a key to improving drought adaptation [6]. Indeed, a higher TE can contribute to a slower rate of soil moisture depletion, and thus leave a higher total soil water content for the grain filling stage [7], a phase that is critical for the drought adaptation of this crop [8]. It has been shown indeed that a higher yield under terminal stress correlated to more water extracted during the grain filling stage [7]. A recent review shows that water availability is critical during key crop stages, especially grain filling and reproductive structure formation, and this is generic across crops facing terminal drought stress [6]. This trait is known to be a complex term that is dependent, on not only physiological factors but also environmental variables [9]. Given this, it seems worthwhile to measure this trait under several environmental conditions and explore the range of genetic variation and putative Genotype x Environment interactions for that trait in a pearl millet diversity panel.

Measuring TE has been the object of a lot of research in the last three or four decades or so, much of it focusing on either transient leaf-based measurement or use of proxies. More recently, a gravimetric method using lysimeters has been developed, where TE is measured as the ratio of biomass accumulated per unit of water transpired over the entire crop cycle [10]. This method enabled the assessment of genetic variation for TE and identify whether there are any significant interactions between genotype and environment. It also allows to evaluate all the components of the Passioura equation (TE, water use, and harvest index [11], and to test their relationships to yield on the same plants using a large set of germplasm [10]. This has been observed previously on sorghum and groundnut collections [12–14]. The precedence of certain components was either dependent on the genetic background of used material [12] or on environmental conditions [14].

A few studies have been carried out in lysimeters with an assessment of TE on pearl millet panel [7, 15–17]. However, the genotypic differences were examined on very limited panels of genotypes. The first objective of our study was to evaluate TE diversity in a larger and more diversified panel, including inbred lines from the PMiGAP collection (Pearl Millet Inbred Germplasm Association Panel), testcross hybrids from this panel, and hybrids targeted for different agroecological zones in India. The second objective was to compare genotypic responses in different environments. Finally, the last objective was to assess the influence of TE on yield across a range of water treatments, environments, and germplasm. Practically, we assessed TE, grain yield and related traits under different conditions: (i) under well-watered and terminal water-stress conditions for the same panel and season, (ii) between two seasons on the same panel and under the same irrigation regime; (iii) between a population of inbreds and hybrids, during the same season and under the same irrigation regime.

## Material and methods

### Plant material and experiments

Three lysimeter experiments were carried out in the International Crops Research Institute for the Semi-Arid Tropics (ICRISAT, Patancheru, Hyderabad, India) in 2012 (Exp. 1), in 2013 (Exp. 2) and 2015 (Exp. 3).

Weather data was recorded during the trial periods: minimum temperatures were 12.2˚C, 11˚C and 12.8˚C in 2012, 2013 and 2015 respectively, while maximum temperatures were 40.2˚C, 41.2˚C and 43.2˚C. Relative humidity (RH) was measured between 7am and 5pm. The

minimum RH was 35%, 38% and 43% in 2012, 2013 and 2015 respectively, while the maximum RH was 94% in 2012 and 2013, 98% in 2015. The vapor pressure deficit (VPD) was calculated using the method proposed by the FAO. Daily VPD ranged between 2.63 to 6.72 kPa in 2012, 0.66 to 6.73 in 2013 and 0.72 to 7.52 kPa in 2015. The plant material belonged to the PMiGAP panel, a collection of pearl millet germplasm representing most of the genetic diversity in pearl millet (S1 Table). In Exp. 1, 260 test-crosses hybrids of the PMiGAP using 843-22A as a tester were assayed in the lysimeter. In Exp. 2, 70 test-crosses in common with Exp. 1 were studied, and grown under two irrigation regimes, i.e. a fully irrigated treatment (well-watered, WW) and a terminal water-stress (WS). In Exp. 3, 234 PMiGAP inbred lines and 43 hybrids were selected. 40 of these hybrids were bred for three agroecological production zones of India that differ in annual rainfall: i) 14 hybrids developed for the driest production zone of India with less than 400 mm annual rainfall (A1 zone); ii) 13 hybrids developed for intermediate rainfall (A zone); iii) 13 hybrids developed for rainfall above 400 mm (B zone). The boundaries of these breeding target zones had been described previously (Gupta et al., 2013).

## Soil filling and growth conditions in lysimeters

The lysimetric system consisted of a set of cylindrical PVC (PolVinyl Chloride) tubes (25 cm diameter, 2m length). The platform (LysiField) was located outdoor and was equipped with a rainout shelter that could be moved above the crop in case of rain. Each tube was filled with Alfisol, collected from the ICRISAT farm. Alfisol at ICRISAT contain around 40–50% clay, 50–40% sand and about 10% silt. Alfisol was used in the lysimeters because, along with the Vertisol, are the two most prominent soil in Indian agriculture lands and are known for their good to high fertility. The same lysimeters have been used in different experiments and soil was not replaced between trials. For that to happen, the lysimeter platform had been treated as a field and had followed typical cereal-legume rotations. The soil used to fill the lysimeters had been initially sieved, and fertilized with sterilized farm manure (at a rate of 2:50 w:w) and with DAP (DiAmmonium Phosphate) and muriate of potash (both at a rate of 200 mg.kg$^{-1}$) [10]. For each new experiment, DAP (18% N and 46% P) and muriate of potash (60% K) was added at a rate of 2g per lysimeter. Urea (46% N) was added as a top-dressing at a rate of 2 g per lysimeter. The length of lysimeters was designed to be deep enough so that rooting depth would not be limiting in this type of soil.

The tops of the cylinders were equipped with metal collars and chains to allow PVC tubes to be lifted for weighing. Each tube, weighing between 160 and 165 kg, was raised with a block-chained pulley. Between the rings of the cylinder and the pulley, a S-type load cell was inserted (200 kg load capacity and 20 g precision; Mettler-Toledo, Geneva, Switzerland). The same weighing protocol was applied among experiments, following procedures previously described [13]. A few centimeters only separated the lysimeters from one another so that the planting density was close to 10 plants m$^{-2}$, which is typical of pearl millet fields in India. Because of possible border effects, replications were set in parallel to the walls of the trenches. There were five rows of cylinders in each trench, each row representing typically one replication.

All genotypes were sown after wetting the tubes, on 16 February 2012, 14 February 2013, and 18 February 2015, with a rate of 4–5 seeds per hill and 3 hills per tube. The same irrigation method was used for sowing across years. Seedlings were thinned a first time to one seedling per hill at about 10 days after sowing (DAS), and then to one plant per tube at 14 DAS. The three experiments were designed as an alpha lattice. Exp 1 was an Alpha lattice with 4 blocks of 65 genotypes and 5 replications of each accession, and a single well-watered (WW) treatment. Exp 2 design was an Alpha lattice with 2 blocks of 35 genotypes and 5 replications for each

accession for the WS treatment and 2 replications for each accession for the WW treatment. Exp 3 was designed as an Alpha lattice with 4 blocks of 70 genotypes and 5 replications for each accession, and also a single WW treatment.

## Application of irrigation regimes and measurements

In all three experiments, the tubes were kept fully irrigated until the beginning of weighing by regular water addition, i.e. usually 500 mL applied every other day. When the weighing started, irrigation to the WW treatment was done after each weighing (see below for the procedure). The tubes were weighed six times in Exp 1 (27, 35, 42, 49, 57, 82 DAS). In Exp 2, the number of weighing was dependent on the water regime applied: four times in WS condition (21, 32, 48, 81 DAS) and seven times in WW condition (22, 27, 35, 42, 50, 55, and 81 DAS). Finally, the tubes were weighed seven times in Exp 3 (29, 34, 42, 48, 55, 69, 94 DAS). Prior to the first weighing of the lysimeters, these were abundantly watered and left to drain for two nights and one day to reach field capacity. Also, prior to weighing, a plastic sheet and a 2-cm layer of beads was applied on top of the tubes to limit soil evaporation. The first lysimeter weight then represented the field capacity weight and was used as a benchmark for re-watering of the WW condition. Lysimeters of the WW treatment of all three experiments were watered after each weighing to bring back the cylinder weight to field capacity weight minus 2kg to avoid drainage. Each cylinder having an estimated 12–13 L of available soil water, re-watering was then done up to about 85% field capacity. In Exp 2, the tubes of the WS treatment were treated as the WW treatment up until two weighings. All WS tubes received a last irrigation of 3L on 28[th] March (42 DAS) and received no more water after that. Transpiration values were calculated for each cylinder between two weighing measurements, by subtracting the weight measured at the last and the current weight, minus the added water.

## Harvest procedure and statistical analysis

Plants were grown to maturity in all three trials and harvested at the last weighing. The entire aerial part of the plant (grain and forage) was harvested and dried for 3 days in a forced-air oven at 70˚C (Fisher Scientific, Waltham, MA, USA). Stem, leaf, panicle, and grain weights were measured separately for each cylinder. The total dry weight (TOTDW, g biomass) was obtained by adding the weights of leaves, stems, and panicle. In these three experiments, the roots were not harvested (in order to not disturb the soil profile of the lysimeters), so that the estimation of total dry weight was based on the shoot part only. However, it was previously discussed that the root part would likely not affect the TE comparison among genotypes in this lysimetric system, due to a lack of relationship between TE and total water extraction, used as proxy of root mass in previous trials [13]. In addition, root and shoot mass are closely related, so that including root mass in the TE calculation would have, at a minimum, maintained the genotypic ranking for TE and possibly could have increased the reported difference. At the end of the experiment, the sum of transpiration (SUMTR, kg water used) was determined by summing the transpiration values measured every week over the entire crop cycle. Transpiration efficiency (TE, g.kg$^{-1}$) was estimated as the ratio of total dry weight to the sum of transpiration values. The harvest index (HI) was calculated by dividing the grain weight (g) to total biomass (g). The grain yield was estimated as the grain weight per plant (g. plant$^{-1}$). Datasets were checked for outlier measurements. Data points were deleted if their values were outside the range of two times the standard deviation around the grand mean. We described the variables grouped by year, irrigation regime and material type (testcrosses, inbreds, hybrids). Data were analyzed by one-way ANOVA on each group to visualize TE variation among genotypes under the same conditions. The results of this ANOVA were used to calculate the broad-sense

heritability of TE in each condition using the following formula $H^2$ = (Sum Sq)/(Sum Sq +(Residuals/number of replicates)) to obtain an estimation of the phenotypic variance attributable to an overall genetic variance. We used a two-way ANOVA in two scenarios: (i) on common genotypes in the same year (Exp 2) with two irrigation regimes to analyze the G effect and the Genotype x Irrigation treatment interaction effect; and (ii) on common genotypes with the same water treatment in two seasons (Exp 1 and Exp 2) to assess the G effect and the Genotype x Year interaction effect. As the inbreds and testcross hybrids (Exp 3) had no accessions in common, this analysis could not be carried out. Finally, regression analyses and Pearson correlation tests were carried out in three scenarios (i) between the TE component variables within each group, i.e. dry weight, and transpiration within the same year or irrigation (ii) between TE measurements across different years or irrigations on common material and (iii) and between grain yield and the yield component according to the Passioura equation (water use, harvest index and TE [11].

Raw data are available at https://doi.org/10.23708/2VJ0TQ.

## Results

### TE varied among genotypes and was heritable

By and large, all traits measured (dry weight, transpiration, TE, HI, grain yield), showed significant differences among genotypes during Exp 1 in 2012. TE was heritable in these conditions ($H^2$ = 0.68, Table 1). Dry weight and transpiration were strongly correlated with each other (R = 0.89, p-value< 2.2e-16, Fig 1a). During Exp 2 in 2013, most traits showed no difference among genotypes under WW regime, TE and grain yield included (Table 1). Yet, there were only two repetitions to test this condition, compared with five repetitions in the other cases. The TE components under WW regime showed the same strong correlation as in Exp 1 (R = 0.83, p-value < 2.2e-16 for 2013, Fig 1b). For the same year under the WS regime, the TE differences observed among genotypes were significant, and the trait was also greatly heritable ($H^2$ = 0.68, Table 1). The correlation between dry weight and transpiration was weaker under WS (R = 0.44, p-value = 1.273e-4, Fig 1c) than under WW regime. During the Exp 3 in 2015, significant differences among genotypes were observed for all traits in hybrid and inbred panels, except HI which was not significant in the hybrid group. Generally, differences among genotypes were more pronounced in the hybrid group than in the inbred group. TE was highly heritable in both hybrid and inbred panels ($H^2$ = 0.74 for inbreds, $H^2$ = 0.73 for hybrids, Table 1). Biomass production and transpiration were highly correlated for both scenarios, with R = 0.76 (p-value = 1.08e-06) for hybrids and R = 0.67 (p-value < 2.2e-16) for inbred lines (Fig 1d and 1e). More precisely, significant differences in TE were observed among hybrids developed according to their targeted agroecological zones. The TE gradient among these hybrid groups followed the same trend as annual rainfall: the drier the agroecological zone, the greater the evaporative demand, the lower the TE. In addition, genotypic differences were observed among individuals from the same zone in the driest zones A1 (p-value = 0.01), while wetter zones A and B showed no significant TE differences among genotypes (Fig 2).

### TE varied slightly between years and irrigation regimes

Focusing on the impact of years on TE variation, we observed that, all genotypes combined, total dry weight per plant was almost twice as high in 2013 (123 g. plant$^{-1}$) as in 2012 (65 g. plant$^{-1}$). Total transpiration was only 41% higher (38.6 kg in 2013 vs. 25.8 kg in 2012). Overall transpiration efficiency was therefore 1.3 times higher in 2013 than in 2012 (3.22 g.kg$^{-1}$ vs 2.55 g.kg$^{-1}$). Grain yield followed the same trend as TE, with a four-fold higher value in 2013 (42.2 g. plant$^{-1}$) than in 2012 (9.9 g. plant$^{-1}$, Table 1). It was in line with the important year effect

**Table 1. One way ANOVA for the main traits in each experiment.**

| | Experiment<br>Irrigation<br>Plant material | Exp. 1<br>WW<br>Test-cross | Exp. 2<br>WW<br>Test-cross | Exp. 2<br>WS<br>Test-cross | Exp. 3<br>WW<br>Inbred | Exp. 3<br>WW<br>Hybrid |
|---|---|---|---|---|---|---|
| SUMTR | Mean | 25.8 | 38.58 | 18.84 | 25.9 | 26.99 |
| | Min | 12.55 | 19.44 | 13.31 | 12.12 | 17.12 |
| | Max | 36.78 | 54.84 | 23.94 | 38.5 | 36.18 |
| | F-value | 3.533 | 1.471 | 1.308 | 2.123 | 2.026 |
| | Pr(>F) | < 2.2E-16 *** | ns | 7.22E-02 . | 6.06E-14 *** | 1.13E-03 ** |
| | H$^2$ | 0.84 | ns | ns | 0.77 | 0.75 |
| TOTDW | Mean | 65.25 | 123.31 | 66.24 | 72.01 | 79.49 |
| | Min | 34.56 | 59.7 | 42.9 | 24.18 | 30.61 |
| | Max | 92.49 | 184.10 | 89.18 | 117.67 | 127.63 |
| | F-value | 2.652 | 1.417 | 1.363 | 2.233 | 2.423 |
| | Pr(>F) | <2.20E-16 *** | ns | 4.59E-02 *** | 1.02E-15 *** | 5.63E-05 *** |
| | H$^2$ | 0.80 | ns | 0.66 | 0.79 | 0.78 |
| TE | Mean | 2.55 | 3.22 | 3.54 | 2.79 | 2.95 |
| | Min | 1.73 | 2.26 | 2.24 | 1.27 | 1.53 |
| | Max | 3.49 | 4.51 | 4.79 | 4.39 | 4.31 |
| | F-value | 1.388 | 0.883 | 1.482 | 1.693 | 1.841 |
| | Pr(>F) | 4.98E-04 *** | ns | 1.60E-02 * | 1.57E-07 *** | 4.30E-03 ** |
| | H$^2$ | 0.68 | ns | 0.68 | 0.74 | 0.73 |
| HI | Mean | 0.153 | 0.354 | 0.218 | 0.2 | 0.347 |
| | Min | 0.001 | 0.150 | 0.021 | 0.001 | 0.077 |
| | Max | 0.328 | 0.580 | 0.406 | 0.436 | 0.546 |
| | F-value | 1.8735 | 2.0932 | 3.0628 | 3.3977 | 1.1033 |
| | Pr(>F) | 6.13E-11 *** | 0.009 ** | 2.00E-10 *** | <2.20E-16 *** | Ns |
| | H$^2$ | 0.74 | 0.79 | 0.83 | 0.85 | ns |
| GRAIN YIELD | Mean | 9.91 | 42.24 | 7.81 | 14.57 | 28.7 |
| | Min | 0.08 | 6.2 | 1.03 | 0.1 | 3.62 |
| | Max | 26.31 | 84.96 | 18.69 | 48.03 | 59.99 |
| | F-value | 1.688 | 1.484 | 2.4534 | 2.4905 | 1.8096 |
| | Pr(>F) | 4.918e-08 *** | ns | 3.52E-07 *** | <2.2e-16 *** | 0.00537 ** |
| | H$^2$ | 0.72 | ns | 0.80 | 0.80 | 0.73 |

Mean, minimal and maximal values, and one-way ANOVA table showing F-value, p-values and broad sense heritability H$^2$ for sum of transpiration (SUMTR, kg), total dry weight (TOTDW, g), transpiration efficiency (TE, g.kg$^{-1}$), harvest index (HI), and grain yield (g. plant-1) grouped by irrigation applied (well-watered WW or water stressed WS), by year of experiment (Exp 1–2012, Exp 2–2013, and Exp 3–2015) and by type of germplasm used (test-cross, inbreds, hybrids). Symbols *, **, and *** denote significance at p<0.05, p<0.01, and p<0.0001, respectively. Acronym 'ns' means non-significant.

observed (F-value = 354, p-value<2.20E-16, Table 2). The variability of weather conditions could potentially be a source of explanation for the difference observed between Exp 1 and Exp 2. Minimum and maximum temperatures were similar in both years (Fig 3a). However, RH was higher in 2013 for 73% of the duration of the experiment, or 60 days out of the 82-day trial period. This overall higher RH was associated with precipitation-induced humidity peaks during the 2013 field experiment, whereas these peaks were absent in 2012 (Fig 3b). This variability resulted in a higher VPD in 2012 than in 2013, over 67% of the duration of the experiment, created more restrictive conditions. In the meantime, mean TE per genotype were correlated across years (R = 0.25, p = 4,76e-2, Fig 4a), which also reflected the large Genotype x Year interaction for TE (F-value = 1.5045, p = 0.0124, Table 2).

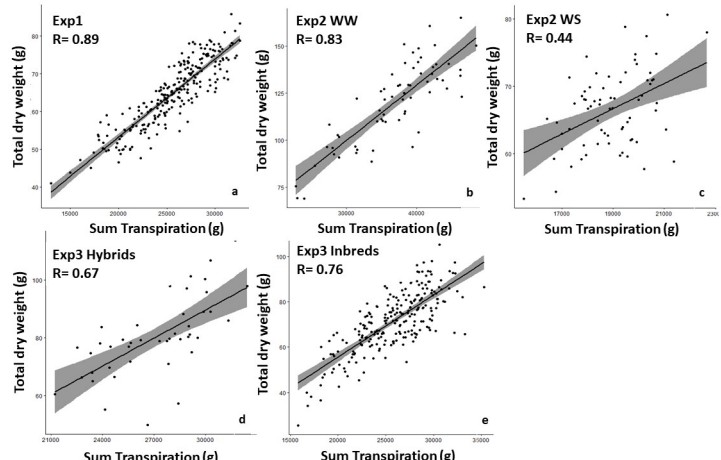

**Fig 1. Relationship between total dry weight (g biomass) and sum of transpiration (g water).** a. with 260 germplasm genotypes in Exp 1 under well-watered conditions, b. with 70 genotypes in Exp 2 under well-watered conditions and c. with 70 genotypes in Exp 2 under water-stressed conditions, d. 43 hybrids in Exp 3 and with e. 234 inbred lines in Exp 3. Data are the mean of five replicated (a,c,d,e) or two replicated lysimeter-grown plants per genotype (b).

Regarding the irrigation regimes applied in 2013, total dry weight production per plant was two-fold higher under WW than under the WS regime, all genotypes combined (123 g. plant$^{-1}$ versus 66 g. plant$^{-1}$). Slightly more than twice as much water was needed to produce biomass under WW conditions than under WS conditions, resulting in a mean TE that was only slightly higher (7.5%) under WS than under WW (3.54 and 3.22 g.kg$^{-1}$ respectively). This was reflected by the large irrigation effect observed in Table 2 (F-value = 38.06, p-value = 1.538e-09). As expected, grain yield was higher under WW (42.2 g. plant-1 versus 7.8 g. plant-1, Table 1). Mean TE of each entry was not correlated across irrigation regimes (R = -0.096, p-value = 0.81, Fig 4b). On the other hand, the Genotype and Genotype x Irrigation effects were comparable (F-value(G) = 1.2557, F-value(GxI) = 0.9038, p-value>0.05).

Concerning the type of germplasm used in Exp 3, the overall total dry weight was higher for hybrid (mean 79.5 g. plant$^{-1}$) than for inbred lines (mean 72.0 g. plant$^{-1}$). This higher biomass

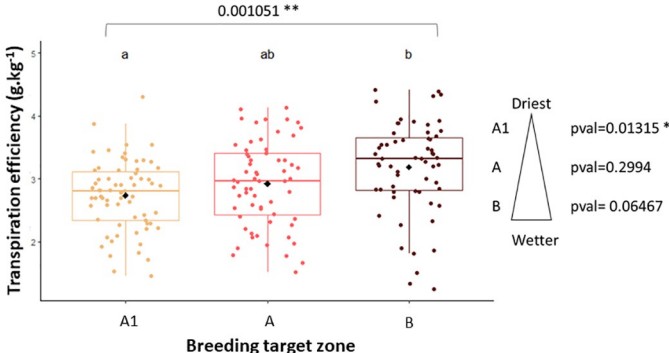

**Fig 2. Transpiration Efficiency (TE, in g. kg$^{-1}$) in different hybrid groups.** TE was measured under well-watered conditions for hybrid groups bred for three different target zones: A1 is the driest zone with less than 400mm of annual rainfall (14 hybrids), A is the intermediate annual rainfall zone (13 hybrids), B is the most humid zone, with an annual rainfall above 400 mm (13 hybrids). Data are the mean of five replicated lysimeter-grown plants per genotype.

**Table 2. Two-way ANOVA summary table for Experiments 1 and 2.**

|  |  | Year (Exp 1 WW/Exp 2 WW) | Irrigation (Exp 2 WW/ Exp 2 WS) |
|---|---|---|---|
| G | F-value | 0.5932 | 1.2557 |
|  | Pr(>F) | ns. | ns. |
| T | F-value | 354.08 | 38.06 |
|  | Pr(>F) | <2.20E-16 *** | 1.538e-09 *** |
| G x T | F-value | 1.5045 | 0.9038 |
|  | Pr(>F) | 0.0124 * | ns. |

The table presents F-value and p-values for genotype effect (G), treatment (T), corresponding to irrigation or year effect (Y), and the Genotype x Treatment, corresponding to Genotype x Year(GxY)) or Genotype x Irrigation (GxI) interaction effect on transpiration efficiency (TE). Year effect was tested with data of Experiments 1 and 2. Irrigation effects were tested in Experiment 2 with two treatments (well-watered conditions WW, watered stressed conditions WS).

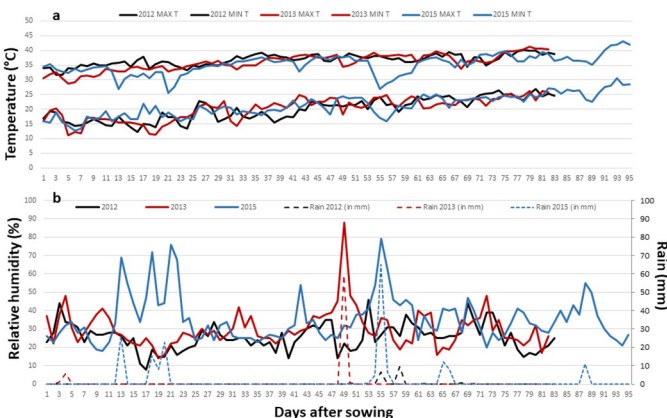

**Fig 3. Environmental conditions (temperature, relative humidity and rain) during the different trials.** (a) Minimum and maximum temperature (˚C), (b) Relative humidity (%) and rain (mm) measured by the local weather station in Patancheru, India, during the trial period of the three experiments (Exp 1 2012, Exp 2 2013, and Exp 3 2015). The trial period is defined in days after sowing. The sowing date were 16 February 2012, 14 February 2013, and 18 February 2015.

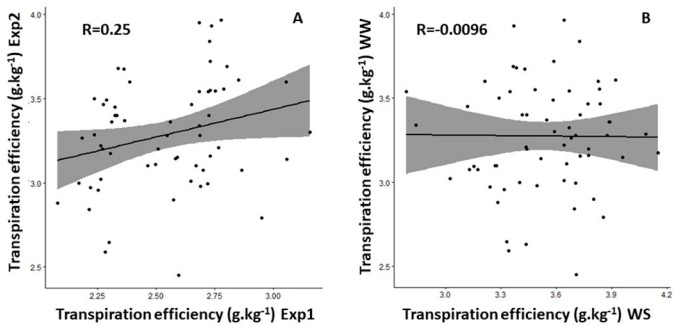

**Fig 4. TE relationships across years and water treatments.** Relationship between transpiration efficiency (g.. kg$^{-1}$) a) in Exp 1 and Exp 2 under well-watered conditions on 70 common genotypes, and b) in Exp 2 under well-watered (WS) and watered-stressed conditions (WW). Data are the means of five replicated (in Exp 1 and Exp 2 WS) or two replicated lysimeter-grown plants per genotype (Exp 2 WW).

production was combined with higher transpiration for hybrids (27.0 kg) than inbreds (25.9 kg). This resulted in a higher TE for hybrids group (2.95 g. plant$^{-1}$) than for inbreds (2.79 g. plant$^{-1}$), and a higher grain yield that was on average doubled for the hybrid group (28.7 g. plant$^{-1}$) compared with inbreds (14.6 g. plant$^{-1}$, Table 1). However, these observations could be linked to a few high-impact individuals that affects the comparison of the whole panel. Indeed, a T-test showed that the mean TE of the 20 best hybrids was 3.90 g.kg$^{-1}$, i.e. significantly lower than the mean TE of the 20 best inbreds (4.11 g.kg$^{-1}$, P< 0.001).

### TE was correlated with the residual yield unexplained by HI

As expected, HI was the main factor explaining yield from the Passioura equation [11], owing to a degree of autocorrelation between HI and yield. Grain yield was also significantly related to TE in all studied years, with the same correlation value under WW (R = 0.27, p-value = 5.64e-3) and WS (R = 0.25, p-value = 1.69e-05) in Exp 2 as well as in Exp 1 (R = 0.30, p-value < 2.2e-16, Table 3). The correlation was much stronger during the Exp 3, with R = 0.45 (p-value < 2.2e-16) on inbred population and R = 0.67 (p-value < 2.2e-16) on hybrid population. By contrast, grain yield was related to water use in Exp 3 only. To remove the confounding effect of HI, the residuals of the linear relationship between grain yield and HI were computed as the Euclidian distance between observed yield values and predicted values from the regression line. These residuals were then plotted against the other possible explanatory factors from Passioura equation (TE and Water Use) [11]. The residual yield showed a higher correlation with TE (from 0.22 to 0.6, Table 3) than with water use (from -0.33 to 0.42, Table 3), suggesting that TE was the second most important factor explaining grain yield differences, besides HI, while water use came only after and in specific experiments.

## Discussion

In this study, pearl millet material was phenotyped during three seasons in a lysimetric platform for TE variation by following up the water uptake during the whole crop cycle. To our knowledge, our study is the first to report variation for TE in a large set of pearl millet germplasm.

### TE variation among genotypes due to genetic effects

Our results showed a significant variation in TE among genotypes in the panel for most of the cases studied, except under WW conditions in Exp 2. Yet, this WW condition had only two

**Table 3. Regression analysis between grain yield and components of Passioura equation.**

| Experiment | Exp. 1 | Exp. 2 | Exp. 2 | Exp. 3 | Exp. 3 |
|---|---|---|---|---|---|
| Irrigation | WW | WW | WS | WW | WW |
| Plant material | Test-cross | Test-cross | Test-cross | Inbred | Hybrid |
| **Grain yield ~ HI** | 0.9 | 0.79 | 0.93 | 0.89 | 0.87 |
| **Grain yield~ TE** | 0.3 | 0.27 | 0.25 | 0.45 | 0.67 |
| **Grain yield ~ WU** | -0.11 | -0.083 | -0.088 | -0.015 | 0.28 |
| **Residual grain yield unexplained by HI ~ TE** | 0.22 | 0.34 | 0.6 | 0.5 | 0.51 |
| **Residual grain yield unexplained by HI ~WU** | -0.0066 | -0.33 | 0.027 | 0.42 | 0.28 |

Harvest index (HI), transpiration efficiency (TE) and water use (WU) were regressed against grain yield. Then the residual grain yield variations unexplained by harvest index were plotted against TE and WU. The panel was grouped by year (Exp 1, Exp 2, Exp 3), by irrigation applied (well-watered conditions WW or water stress conditions WS) and by type of germplasm used (test-cross, inbreds, hybrids).

repetitions instead of five compared to other experiments. The heritability values suggested that TE variation among individuals is determined by a genetic component in each condition studied (from $H^2 = 0.68$ in Exp 1 and Exp 2 under WS with test-crosses to $H^2 = 74$ in Exp 3 under WW with inbreds). Beggi and colleagues [15] (2015) noted a significant variation in TE among 15 genotypes in the case of phosphorus-poor soil, under well-irrigated and water stressed conditions. Vadez et al [7] (2013), focused on temporal differences in water uptake, reported significant differences in TE among 8 genotypes. The values mentioned here under WS and WW (2.30–4.79 g. $kg^{-1}$ water) were above the range found by [15] (1.23–1.82 g. $kg^{-1}$ water) and that in [7] (2.19–2.95 g. $kg^{-1}$ water), hence this current work reports a higher range of variation, of approximately two folds. Differences in the mean values across trials could be related to differences in environmental conditions (different trial periods and locations). For instance, the trial of [15] was carried out under higher evaporative demand. In relation to these TE variations here, dry weight and total transpiration were not always tightly correlated for some conditions (Fig 1). Transpiration efficiency being the slope of the linear regression between dry weight and total transpiration, a loose correlation highlighted that certain genotypes could reach a high biomass at relatively low water cost, hence had higher TE, than other genotypes.

Variation due to genetic factors was also found in Exp 3 among hybrids that differ genetically in their adaptation zone. Indeed, the group of hybrids adapted for the A1 driest zone showed a variation in TE among genotypes, underlying the importance of selecting the best elements adapted to the target production environment. On the contrary, hybrids adapted to A and B wetter zones showed little or no difference among genotypes: the more arid and restrictive the environment, the greater the differences among genotypes. Curiously, if we focus on the mean TE of hybrids across agroecological zones, we observed that the wetter the zone, the higher the TE. This is surprising, given that TE is often considered as a measure of a genotype's drought adaptation, and we could have expected hybrids from the driest zone to have higher TE. Zone A1 has high VPD conditions and poor, sandy soil [18]. In this scenario, a strategy to conserve water that could easily evaporate may not be suitable. Instead, it is likely that genotypes adapted to this zone may have developed a rather opportunistic strategy of using the available water as quickly as possible before it evaporates, and then banking on high tillering characteristics and small grains to limit the impact of intermittent drought [19]. Some genotypes, such as H77-833/2, are known to adopt this type of strategy [7, 20, 21]. This is also what we found in an earlier study with hybrids from the A1, A, and B zone [22]. In that study, while the slope of the response of transpiration at low VPD value was similar in groups of hybrids from the three zones, there was an inflection in the transpiration response above a VPD of about 2kPa in the group of hybrids from the B zone, i.e. the wet zone. This would imply restricting transpiration under high VPD, which would increase TE [9, 23], in line with our results here. Additional research would then be necessary to confirm these water use strategies, i.e. opportunist versus conservative, in hybrids adapted to dry or wet zones.

Independently of this water use strategy hypothesis, our results show that the notion of TE as a mirror of plant performance under water stress could be revisited to determine whether it is an appropriate method for measuring drought tolerance in pearl millet. Looking at other environmental factors, like the soil, which also have a crop specie dependent effect on TE [24], should also be carefully taken into account. In addition, the characterization of pearl millet production zones as A1, A, B could also be reviewed. These zones have been determined on the basis of annual rainfall in Indian regions and pearl millet growing areas. It has remained unchanged since 1979 and is still considered a reference in most pearl millet breeding programs. Recently, a more advanced characterization based on crop modeling of these production zones has been established [25, 26]. This work has allowed a more precise delimitation of

five new target production environments in the pearl millet growing zone in India. Testing TE variation according to this new zonation would also be a way of refining our inter-zone observations. Finally, the fact that some of the variation in TE appears to be due to important genetic effects highlights the need for further study towards a genome-wide association analysis to determine the genetic basis of TE variation.

## TE variation due to the Genotype x Environment interaction

As expected, we observed a strong environmental effect on TE, whether by year or irrigation. Looking more closely at Exp 2, the TE difference between irrigation treatments was only 7.5%. This was a smaller difference than reported earlier in sorghum where TE under WS was 12.5% higher than under WW conditions in this drought-tolerant C4 crop [13]. Although not significant here, Genotype effect and Genotype x Irrigation interaction showed almost equal F-values, suggesting that both factors had equal importance on TE variation. This genotypic effect was significant in Exp 1 and Exp 2 under WS conditions with five replicates, and insignificant during Exp 2 under WW conditions with two replicates. In this context, we can presume that we didn't capture all TE's existing genetic diversity in Exp 2 under WW conditions, possibly because only two replications were used for the WW treatment in Exp 2, and this lack of significance biased the inter-irrigation analysis.

On the other hand, the Genotype x Year interaction was clearly more significant. As TE depend partially on environmental variables such as the atmospheric vapor pressure deficit, the effect of the year was likely linked to the different meteorological conditions observed over the two years of experimentation.

In summary, TE differences induced by Genotype x Environment interactions occurred in our field-like trials, in particular Genotype x Year interaction. It therefore appears to be an important component to consider in capturing TE variation. Further work is needed on the phenotypic plasticity of TE expressed in different environments in the potential perspective of TE-based selection. However, based on our results here, it would show that Genotype x Environment interactions were still weaker than the differences induced by genotypic effects. This finding is in line with the TE evaluation cited above, where the Genotype x Treatment interaction was not significant in all cases studied [7, 15, 17]. From these initial observations, we can draw one main conclusion for future TE phenotyping on a panel of pearl millet. Repeating the experiment over several years seems essential to consider weather conditions and limit their impact on TE variations, to consider Genotype x Environment interactions in TE assessment.

## TE, the second contributor to grain yield

Whatever the origins of the variation, TE was an important factor to consider for pearl millet yield. After removing the autocorrelation HI effect, TE explained a large portion of the remaining grain yield variations in all conditions studied, highlighting the benefit of selecting germplasm based on high TE for breeding programs. This result is in line with the result found on other crops such as sorghum panel [13], which showed that TE variation in this species depended on the genetic background of the germplasm studied [12]. In this study, the genetic background of the pearl millet genotypes was not analyzed and would be a way of improving our knowledge on the genetic source of TE variation.

In summary, improvement of TE-based selection in pearl millet could be based on several levers of action, such as (i) improve our knowledge on TE plasticity on the new target production environments to gauge the extent of G*E interactions (ii) investigate the genetic basis of TE in pearl millet on the panels presented here and compare to other crops. A comparative

study of sorghum and pearl millet on the genetic basis of TE is currently the objective of additional studies.

## Supporting information

**S1 Table. List of germplasm tested.** The table indicates the year when it was tested, the type of genetic material (Test-cross hybrid, inbred, hybrid), the Generation Challenge Program (GCP) entry number, the test-cross pedigree and the breeding target zone when applicable.
(XLSX)

**S2 Table. Value of traits measured in the different trials.** Besides genotype name, experiment, germplasm types the table reports transpiration, total dry weight, TE, harvest index (HI) and grain yield.
(XLSX)

## Author Contributions

**Conceptualization:** Charles Thomas Hash, Vincent Vadez.

**Data curation:** Laura Grégoire, Vincent Vadez.

**Formal analysis:** Laura Grégoire, Yves Vigouroux, Vincent Vadez.

**Funding acquisition:** Vincent Vadez.

**Investigation:** Vincent Vadez.

**Methodology:** Jana Kholova, Vincent Vadez.

**Project administration:** Vincent Vadez.

**Resources:** Rakesh Srivastava, Charles Thomas Hash, Vincent Vadez.

**Supervision:** Yves Vigouroux, Vincent Vadez.

**Writing – original draft:** Laura Grégoire.

**Writing – review & editing:** Jana Kholova, Yves Vigouroux, Vincent Vadez.

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
