## [Decision Letter · Decision Letter 0]

8 May 2024

PONE-D-24-06097Transpiration efficiency variations in the pearl millet reference collection PMiGAPPLOS ONE

Dear Dr. Vadez,

Thank you for submitting your manuscript to PLOS ONE. After careful consideration, we feel that it has merit but does not fully meet PLOS ONE’s publication criteria as it currently stands. The reviewers recommend reconsideration of your manuscript following minor revision. I invite you to resubmit your manuscript after addressing the comments below.  Therefore, we invite you to submit a revised version of the manuscript that addresses the points raised during the review process. Please note that your revised submission may need to be re-reviewed.  

We look forward to receiving your revised manuscript.

Kind regards,

Somashekhar Mallikarjun Punnuri, PhD

Academic Editor

PLOS ONE

Journal Requirements:

 [Agence Nationale de la Recherche (ANR, grant ANR-17-MPGA-0011)].  

[LG and VV was supported by the Make Our Planet Great Again (MOPGA) ICARUS project (Improve Crops in Arid Regions and future climates) funded by the Agence Nationale de la Recherche (ANR, grant ANR-17-MPGA-0011), by the Occitanie Region through a financial contribution to grant ANR-17-MPGA-0011, and by Montpellier University of Excellence (I-Site MUSE). JK was supported by the Internal grant agency of the Faculty of Economics and Management, Czech University of Life Sciences Prague, grant no. 2023B0005 (Oborově zaměřené datové modely pro podporu iniciativy Open Science a principu FAIR).]

 [Agence Nationale de la Recherche (ANR, grant ANR-17-MPGA-0011)]. 

5. In the online submission form, you indicated that [The data underlying the results presented in the study are available upon request from the corresponding author]. 

Reviewers' comments:

Reviewer's Responses to Questions

**Comments to the Author**

1. Is the manuscript technically sound, and do the data support the conclusions?

Reviewer #1: Yes

Reviewer #2: Yes

Reviewer #3: Yes

2. Has the statistical analysis been performed appropriately and rigorously? 

Reviewer #1: Yes

Reviewer #2: Yes

Reviewer #3: Yes

3. Have the authors made all data underlying the findings in their manuscript fully available?

Reviewer #1: Yes

Reviewer #2: Yes

Reviewer #3: Yes

4. Is the manuscript presented in an intelligible fashion and written in standard English?

Reviewer #1: Yes

Reviewer #2: Yes

Reviewer #3: Yes

5. Review Comments to the Author

Reviewer #1: The manuscript "Transpiration efficiency variations in the pearl millet reference collection PMiGAP" by Gregoire and colleagues explains about variations in Transpiration Efficiency in pearl millet genotypes under well watered and water stress conditions. This study highlights different source of variation in transpiration and contribution of the trait to grain yield. Overall the manuscript is well written. Data is reliable and sufficient to explain the importance of trait under well water and water stress conditions. The data is important for further studies such as association mapping to reveal genetic basis of trait. I enjoyed reading the manuscript. I recommend it for publication, provided some minor corrections may be incorporated in the current version. Figure are good quality. Typos at some places may be corrected in the revised version.

1. Line 30-32: Sentence is incomplete

2. Line 43-45: Sentence looks in contrast to Line 39.

3. Transpiration is well known avoidance strategy in plants. I think avoidance should be kept separate from tissue level tolerance where plant actually make change at the whole plant level to combat with stress condition. Hence, I suggest avoidance is better choice instead of tolerance

4. Line 166-168: For the better understanding of readers, it can be explained little more why genetic variation in root biomass would not impact transpiration efficiency. Also, its more relevant between WW and WS treatments.

5. Line 344-352: This is very interesting observation. Is there any data in this direction for example transpiration rate (other studies) which can be cited here?

6. Line 358-360: I think drought avoidance is a complex process and highly plastic. Also, plants may change their strategy depending upon the environment or timing and severity of stress. As a part of drought tolerance, drought avoidance through TE could be a valid strategy. However, root system architecture, soil texture would be major players to decide that (which is not part of this study).

Reviewer #2: The study is indeed one of its kind for TE work in Pearl millet. I recommend publication with minor revisions.

Please see below some minor comments for the authors:

Line 47 – Change “in” to “to”

Line 71 – …our study was then to.. – remove “then”

Line 109 – unit for grain yield should be g.plant-1

Line 114- …lysimetric system consisted in.. – replace “in” with “of”

Line 121 – specify what “DAP” stands for

Line 137 – specify what DAS stands for. All acronym should be explained as it full form when they appear for the first time

Line 185 – space between Exp 3

Line 319 – …a significant variation of… - replace “of” with “in”

Line 326 – add “ed” to stress – stressed

Line 361 – replace “could” with “should”

Line 377 – consider using the same acronyms. For example, Genotype x Irrigation interaction

Line 383 – should read “ was clearly more significant”

Line 384 – should read CO2.

Line 384 – provide evidence of differences in atmospheric CO2 to show differences between the years

Line 404 - …other crops such as sorghum panel…” connect the two sentences using “which showed that sorghum…”

Line 406 – Consider using.. “In this study…” the genetic background of….

Line 407 - ….”would be a way of improving out knowledge” about what?

Line 412 – “…currently the object…” Replace “object” with “objective”

Line 422 – Replace “was” with “were”

Line 485, 488 – Reference missing DOI – Be consistent in your referencing style

Reviewer #3: Grégoire et al. investigate transpiration efficiency (TE) in pearl millet, which is crucial for crop performance under water-limited conditions. Analyzing a diverse panel, including inbred lines and hybrids, they uncover significant variability in TE across different environments. TE emerges as a key determinant of grain yield, highlighting its importance in breeding programs. The study identifies promising avenues for future research, enhancing our understanding of pearl millet's resilience to water stress. Overall, this study provides valuable insights into pearl millet TE and its implications for breeding in water-stressed environments. However, there are areas that could be strengthened by providing additional justification and clarity before considering the manuscript for publication.

Lines 39-43: The introduction effectively establishes the importance of pearl millet as a drought-tolerant cereal however, it would enhance clarity to provide a brief definition or explanation of terminal drought stress for readers who may not be familiar with the term.

Lines 43-44: While stating that 'This abiotic stress has the most radical impact on pearl millet grain and forage yield, as well as yield stability,' it would strengthen the argument to quantify the extent of yield loss associated with this stress. Providing specific figures or percentages would enhance the reader's understanding of the magnitude of the impact.

Lines 46-49: The rationale for investigating TE as a key trait for improving drought tolerance in pearl millet is well-articulated. However, consider providing a bit more context on why the grain-filling stage is critical for drought tolerance in pearl millet to further emphasize the significance of TE in this context.

Line 114-121: Please provide the full form of the abbreviation used in the materials and method section for example PVC, DAP, DAS…….etc.

Line 116: “Alfisol” it would be beneficial to mention about what type of soil is alfisol, if it has any specific soil particle composition that is important for TE study????

Line 121: “DAP and MOP” , clearly mention the composition of NPK on them like you mentioned for Urea.

Line 138-142: Describe clearly about the water conditions in Exp 1 and Exp 3 like you did for Exp 2. If all genotyped planted in same water regimes, mention that as well. Was there any type of control in any of the experiments??If yes please mention them as well.

Line 162-163: “Forced air oven at 70 degree C” please mention the type , make and manufacturer of the oven used.

Line 170: Please be consistent with the unit used for TE (TE, g.kg-1) or (g. kg–1 water) or (g. biomass. kg water -1)

Line 195-198: Be consistent “among genotypes” or “among entires”

Line 112: “Acronym ‘ns’ means non-significant” mention similarly for NA

Line 247: “which also reflected the large GxYear interaction for TE (F-value = 1.5045, p= 0.0124, Table 2)”, I did not see GXYear in the table 2. If it is same as GXT, mention it clearly in the footnote.

Line 265-267: This result is not clear to me “Slightly more than twice as much water was needed to produce biomass under WW conditions than under WS conditions, resulting in a mean TE that was only slightly higher (7.5%) under WS than under WW (3.54 and 3.22 g.kg-1 respectively)”. Despite a significant difference in water use between well-watered (WW) and water-stressed (WS) conditions, the mean TE was only slightly higher under WS compared to WW. This unexpected result contradicts the typical understanding that water-stressed conditions should lead to higher TE due to plants' physiological responses to conserve water. Additionally, the lack of correlation between TE and irrigation regimes further complicates the interpretation of the data. These inconsistencies raise questions about the accuracy of the measurements or the validity of the experimental setup, requiring further investigation and clarification.

Line 291-384: The issue here lies in the interpretation of the correlation between transpiration efficiency (TE) and the residual yield unexplained by harvest index (HI). While the correlation values suggest a relationship between TE and yield, attributing causality solely to TE might be misleading. The statement implies that TE directly affects yield beyond the influence of HI, but it overlooks potential confounding factors or alternative explanations for the observed correlations.

Fig 2: Supplementary table 2 shows the number of lines in hybrid group (14A1,14A and 12B) different than in supplementary table 1 (14A1,13A and 13B. Which data was used for this figure write correct numbers of line used.

6. PLOS authors have the option to publish the peer review history of their article (what does this mean?). If published, this will include your full peer review and any attached files.

Reviewer #1: **Yes: **Rajeev Nayan Bahuguna

Reviewer #2: No

Reviewer #3: No

---

## [Author Response · Author response to Decision Letter 0]

17 Jun 2024

Journal Requirements:

Answer: We believe we have followed these recommendations to the best of our understanding of these two guideline documents

Answer: We included that information

 [Agence Nationale de la Recherche (ANR, grant ANR-17-MPGA-0011)]. 

Answer: We have indeed added this statement in the Acknowledgement section of the manuscript. 

[LG and VV was supported by the Make Our Planet Great Again (MOPGA) ICARUS project (Improve Crops in Arid Regions and future climates) funded by the Agence Nationale de la Recherche (ANR, grant ANR-17-MPGA-0011), by the Occitanie Region through a financial contribution to grant ANR-17-MPGA-0011, and by Montpellier University of Excellence (I-Site MUSE). JK was supported by the Internal grant agency of the Faculty of Economics and Management, Czech University of Life Sciences Prague, grant no. 2023B0005 (Oborově zaměřené datové modely pro podporu iniciativy Open Science a principu FAIR).]

 [Agence Nationale de la Recherche (ANR, grant ANR-17-MPGA-0011)]. 

Answer: I have to admit that we don’t fully understand what we need to do. For us, it is compulsory to have funding sources acknowledged in the text of the manuscript and this is usually done in the Acknowledgement section. Therefore, please modify the Funding Statement as follows:

LG and VV was supported by the Make Our Planet Great Again (MOPGA) ICARUS project (Improve Crops in Arid Regions and future climates) funded by the Agence Nationale de la Recherche (ANR, grant ANR-17-MPGA-0011), by the Occitanie Region through a financial contribution to grant ANR-17-MPGA-0011, and by Montpellier University of Excellence (I-Site MUSE). JK was supported by the Internal grant agency of the Faculty of Economics and Management, Czech University of Life Sciences Prague, grant no. 2023B0005 (Oborově zaměřené datové modely pro podporu iniciativy Open Science a principu FAIR).

5. In the online submission form, you indicated that [The data underlying the results presented in the study are available upon request from the corresponding author]. 

Answer: Data are in Supplementary Table 2. A deposit of the raw date has been made on datasuds and the DOI is "Transpiration efficiency variations in the pearl millet reference collection PMiGAP", https://doi.org/10.23708/2VJ0TQ, DataSuds. We have made mention of it in the manuscript.

Answer: No issue with this, the reference list has been completed and correct. No retracted papers have been cited.

Reviewers' comments:

Reviewer's Responses to Questions

Comments to the Author

1. Is the manuscript technically sound, and do the data support the conclusions?

Reviewer #1: Yes

Reviewer #2: Yes

Reviewer #3: Yes

2. Has the statistical analysis been performed appropriately and rigorously?

Reviewer #1: Yes

Reviewer #2: Yes

Reviewer #3: Yes

3. Have the authors made all data underlying the findings in their manuscript fully available?

Reviewer #1: Yes

Reviewer #2: Yes

Reviewer #3: Yes

4. Is the manuscript presented in an intelligible fashion and written in standard English?

Reviewer #1: Yes

Reviewer #2: Yes

Reviewer #3: Yes

5. Review Comments to the Author

Reviewer #1: The manuscript "Transpiration efficiency variations in the pearl millet reference collection PMiGAP" by Gregoire and colleagues explains about variations in Transpiration Efficiency in pearl millet genotypes under well watered and water stress conditions. This study highlights different source of variation in transpiration and contribution of the trait to grain yield. Overall the manuscript is well written. Data is reliable and sufficient to explain the importance of trait under well water and water stress conditions. The data is important for further studies such as association mapping to reveal genetic basis of trait. I enjoyed reading the manuscript. I recommend it for publication, provided some minor corrections may be incorporated in the current version. Figure are good quality. Typos at some places may be corrected in the revised version.

Answer: We thank the reviewer of these very positive comments. We have addressed the suggestions in the revised manuscript

1. Line 30-32: Sentence is incomplete

Answer: The sentence has been changed to: “Transpiration efficiency was the second contributor to grain yield after harvest index, highlighting the importance of integrating it into pearl millet breeding programs »

2. Line 43-45: Sentence looks in contrast to Line 39.

Answer: We wanted to emphasize that, even though pearl millet is a crop that is particularly well adapted to drought conditions compared to other cereals, it is still subject to abiotic stress, particularly terminal water stress. The sentence has been changed to: “Despite its drought tolerance, this cereal is nonetheless subject to abiotic stresses, such as terminal drought stress »

3. Transpiration is well known avoidance strategy in plants. I think avoidance should be kept separate from tissue level tolerance where plant actually make change at the whole plant level to combat with stress condition. Hence, I suggest avoidance is better choice instead of tolerance

Answer: We thank the reviewer for this comment. We agree that the term “tolerance” is misleading. We actually cited a recent review where we have re-visited these terms. On the basis of that recent review (2024), we have replaced with “adaptation”.

4. Line 166-168: For the better understanding of readers, it can be explained little more why genetic variation in root biomass would not impact transpiration efficiency. Also, its more relevant between WW and WS treatments.

Answer: We did not say that not including the root mass would not affect TE – of course it would because plant mass would increase if we added the root mass. What we said is that not including the root mass for the calculation of TE would “not affect the TE comparison among genotypes”. We added that, in previous experiments on the same lysimeter, it was demonstrated that TE was not correlated to the total water extraction, which was used as a proxy of root mass. We have also refered to the close relationship between shoot and root mass in plant. A consequence is that the ranking of TE values would not be modified if we included root mass. Hence, the analysis of TE comparison among genotypes should not be impacted by the fact we did not include root biomass. We have also added another important reason for not including the root mass, which was because we did not want to disturb the soil profile of these lysimeters.

5. Line 344-352: This is very interesting observation. Is there any data in this direction for example transpiration rate (other studies) which can be cited here?

Answer: We are glad that the reviewer picked up that part of the discussion, because it rather goes against the common understanding of drought adaptation in crops (save water rather than spend). Well, all is context dependent and this strategy seems to make sense for the case of pearl millet. We cited Kholova et al 2010 and Vadez et al 2013. In these works, we report data on transpiration rates being higher in lines like H77-833/2, that are those that are adapted to the A1 zone (the dry zone). There are not much data on that unfortunately although it reminded me that we published a paper a few years ago (Medina et al 2017) where indeed we found that hybrids from the wet zone (B) did restrict transpiration under high VPD, which can be linked up to possible increase in TE according to Sinclair et al 2005. We included that in the discussion, along with Medina et al paper. And tks again for inducing this important addition.

6. Line 358-360: I think drought avoidance is a complex process and highly plastic. Also, plants may change their strategy depending upon the environment or timing and severity of stress. As a part of drought tolerance, drought avoidance through TE could be a valid strategy. However, root system architecture, soil texture would be major players to decide that (which is not part of this study).

Answer: We fully agree with you that water use strategy depends on many parameters, including soil and root system parameters. In fact, we have added a reference that point to that very same issue and where we compared TE in different crop species in different soils. Comparing TE among genotypes remains possible on genotypes of the same year, under the same irrigation, because they have evolved in the same environment, including soil and atmosphere parameters. However, we can see here the plasticity of the trait when we focus on inter-year or inter-treatment comparisons. Hence, in our opinion, the importance of continuing the study on trait plasticity, by investigating the causes of GxE.

Reviewer #2: The study is indeed one of its kind for TE work in Pearl millet. I recommend publication with minor revisions.

Answer: We thank the reviewer for this very positive comment.

Please see below some minor comments for the authors:

Line 47 – Change “in” to “to”

Answer: Done

Line 71 – …our study was then to.. – remove “then”

Answer: Done

Line 109 – unit for grain yield should be g.plant-1

Answer: done

Line 114- …lysimetric system consisted in.. – replace “in” with “of”

Answer: Done

Line 121 – specify what “DAP” stands for

Answer: Done

Line 137 – specify what DAS stands for. All acronym should be explained as it full form when they appear for the first time

Answer: Done

Line 185 – space between Exp 3 Done

Line 319 – …a significant variation of… - replace “of” with “in”

Answer: Done

Line 326 – add “ed” to stress – stressed Done

Line 361 – replace “could” with “should”

Answer: Done

Line 377 – consider using the same acronyms. For example, Genotype x Irrigation interaction

Answer: Done. The format Genotye x Irrigation, Genotype x Year or Genotype x Environment has been corrected for all occurrences.

Line 383 – should read “ was clearly more significant”

Answer: Done

Line 384 – should read CO2.

Answer: Done

Line 384 – provide evidence of differences in atmospheric CO2 to show differences between the years

Answer: We have removed the mention to possible differences in CO2 across year, because these would not be large enough to have an effect on TE. 

Line 404 - …other crops such as sorghum panel…” connect the two sentences using “which showed that sorghum…”

Answer: Done

Line 406 – Consider using.. “In this study…” the genetic background of….

Answer: Done

Line 407 - ….”would be a way of improving out knowledge” about what?

Answer: ‘In this study, the genetic background of the pearl millet genotypes was not analyzed and would be a way of improving our knowledge on the genetic source of TE variation.”

Line 412 – “…currently the object…” Replace “object” with “objective”

Answer: Done

Line 422 – Replace “was” with “were”

Answer: Done

Line 485, 488 – Reference missing DOI – Be consistent in your referencing style

Answer: Missing DOIs have been added, when available. One reference does not have DOIs (Bidinger et Hash, 2004) 

Reviewer #3: Grégoire et al. investigate transpiration efficiency (TE) in pearl millet, which is crucial for crop performance under water-limited conditions. Analyzing a diverse panel, including inbred lines and hybrids, they uncover significant variability in TE across different environments. TE emerges as a key determinant of grain yield, highlighting its importance in breeding programs. The study identifies promising avenues for future research, enhancing our understanding of pearl millet's r

---

## [Decision Letter · Decision Letter 1]

12 Jul 2024

Transpiration efficiency variations in the pearl millet reference collection PMiGAP

PONE-D-24-06097R1

Dear Dr. Vadez,

We’re pleased to inform you that your manuscript has been judged scientifically suitable for publication and will be formally accepted for publication once it meets all outstanding technical requirements.

Kind regards,

Vanessa Carels

Staff Editor

PLOS ONE

Additional Editor Comments (optional):

Reviewers' comments:

Reviewer's Responses to Questions

**Comments to the Author**

1. If the authors have adequately addressed your comments raised in a previous round of review and you feel that this manuscript is now acceptable for publication, you may indicate that here to bypass the “Comments to the Author” section, enter your conflict of interest statement in the “Confidential to Editor” section, and submit your "Accept" recommendation.

Reviewer #1: All comments have been addressed

Reviewer #3: All comments have been addressed

2. Is the manuscript technically sound, and do the data support the conclusions?

Reviewer #1: Yes

Reviewer #3: Yes

3. Has the statistical analysis been performed appropriately and rigorously? 

Reviewer #1: Yes

Reviewer #3: Yes

4. Have the authors made all data underlying the findings in their manuscript fully available?

Reviewer #1: Yes

Reviewer #3: Yes

5. Is the manuscript presented in an intelligible fashion and written in standard English?

Reviewer #1: Yes

Reviewer #3: Yes

6. Review Comments to the Author

Reviewer #1: (No Response)

Reviewer #3: The authors have addressed all the comments thoroughly. The revisions made have significantly improved the clarity and depth of the manuscript. Each point raised in the initial review has been carefully considered and responded to in detail. I recommend accepting the manuscript for publication.

7. PLOS authors have the option to publish the peer review history of their article (what does this mean?). If published, this will include your full peer review and any attached files.

Reviewer #1: No

Reviewer #3: No

---

## [Editor Report · Acceptance letter]

18 Jul 2024

PONE-D-24-06097R1 

PLOS ONE

Dear Dr. Vadez, 

I'm pleased to inform you that your manuscript has been deemed suitable for publication in PLOS ONE. Congratulations! Your manuscript is now being handed over to our production team.

Kind regards, 

on behalf of

Dr. Vanessa Carels 

Staff Editor

PLOS ONE